# Gray Ramus Communicans Nerve Block for Acute Pain Control in Vertebral Compression Fracture

**DOI:** 10.3390/medicina57080744

**Published:** 2021-07-23

**Authors:** Dou-Young Park, Il Choi, Tae-Gyum Kim, Woo-Jae Kim, Il-Young Shin, Eun-Kyung Khil

**Affiliations:** 1Department of Neurological Surgery, Dongtan Sacred Heart Hospital, Hallym University Hwaseong, Hwaseong 18450, Korea; Dypark1016@gmail.com (D.-Y.P.); kosaken@hallym.or.kr (I.-Y.S.); 2Department of Neurosurgery, Ace Hospital, Ansan 39177, Korea; kimtk0311@hallym.or.kr; 3Department of Neurosurgery, Youngdong Hospital, Hwaseong 38611, Korea; mediboy1@hallym.or.kr; 4Department of Radiology, Dongtan Sacred Heart Hospital, Hallym Unverisity, Hwaseong 18450, Korea; nizzinim@gmail.com

**Keywords:** osteoporosis, compression fracture, bone marrow density, serial visual analog scale, Oswestry Low Back Disability Questionnaire, Roland–Morris Disability Questionnaire

## Abstract

*Background and Objectives*: The current options for acute pain control of vertebral compression fracture include hard brace, vertebroplasty, early surgery, and analgesic injection. We hypothesize that the gray ramus communicans nerve block (GRNB) controls the acute pain experienced during vertebral compression fractures. This study assessed the time course of pain control after injection and evaluated the risk factors affecting pain control failure. *Materials and methods:* Sixty-three patients (24 male, 66.19 ± 15.17 y) with a thoracolumbar vertebral fracture at the T10-L5 spine, who presented to our hospital from November 2018 to October 2019, were included in this retrospective cohort study. GRNB was performed within 1 week of the trauma. The patients were followed up on days 3, 14, 30, 90, and 180 and assessed with the serial visual analog scale (VAS, resting and motion), Oswestry Low Back Disability (ODI) questionnaire, and Roland–Morris Disability Questionnaire (RDQ). The failure group was defined by the need for an additional block or cement injection after a single GRNB. The failure group’s risk factors, such as body mass index, initial thoracolumbar injury classification and severity score, Kummel’s disease, age, bone marrow density (BMD), and underlying disease, were analyzed. *Results*: The motion VAS score improved from preoperative to three months post-procedure, but the resting VAS was affected by the procedure for only three days. The quality of life index improved at postoperative six months. A lower BMD was the only risk that affected treatment failure in the logistic regression analysis (*p* = 0.0038). *Conclusion*: The effect of GRNB was maintained even at three months after trauma based on motion VAS results. The only risk factor identified for GRNB failure was lower BMD.

## 1. Introduction

As we have entered an aging society, osteoporotic vertebral fractures are on the rise, and the number of patients with traumatic vertebral fractures is increasing [1]. The prevalence of osteoporosis has been reported as 40–90% in women >50 years. Spinal compression fracture is the most common complication, and a 12–40% prevalence of osteoporotic compression fractures has been reported among individuals >50 years [2,3,4]. Most vertebral fractures heal naturally over time, and the pain is relieved. However, in some patients, especially in the elderly patient group, many patients suffer from intolerable pain, neurologic deficits, and, rarely, deformities, which lead to a decreased quality of life. The patients with spine fractures suffered from pain which prolongs hospitalization and increases the bed rest period. Therefore, the pain interventions that effectively manage pain and shorten recovery time would be of great benefit [5,6]. Several papers have demonstrated that vertebroplasty or kyphoplasty effectively relieves pain in patients with osteoporotic vertebral fracture [6,7,8,9]. As an alternative, there have been studies on gray ramus nerve block (GRNB), which has been shown to affect pain relief [10,11,12,13,14]. However, there has been no study on how long the effect of gray ramus nerve block was maintained and what patient characteristic impacts its duration. Therefore, we designed this study to confirm that the GNRB effectively reduces pain in patients with vertebral body fractures and to analyze the cause of treatment failure in the ineffective group.

The primary goal of this study was to analyze the time course of pain control after GNRB. Secondary goals were to evaluate the risk factors for failing pain control after GNRB. We defined ‘pain control failure’ as requiring additional procedures such as a repeated block or vertebroplasty or kyphoplasty.

## 2. Materials and Methods

### 2.1. Study Design, Setting, and Patient Population

The Institutional Review Board at Hallym University reviewed and approved this retrospective study. We found spinal fracture patients admitted to the neurosurgery department between 1 November 2018 and 1 November 2019 through electronic medical record (EMR) system. As inclusion criteria, patients had an index thoracolumbar fracture from T10 to L5, adult patients over 18 years of age were included, and patients who underwent computed tomography (CT)or magnetic resonance imaging (MRI) for fracture evaluation were included. The exclusion criteria were as follows: visual analog scale (VAS) score was 3 or less; no acute fracture on CT or MRI; spine fracture which needs fixation surgery; spine fractures other than T10-L5 fractures; multiple trauma combined T10-L5 fractures; pathologic fractures such as tumor, infection, etc.; severe soft tissue injury such as multiple burn injuries (Figure 1).

For all patients, thoraco-lumbo-sacral orthosis braces were worn, and conservative treatment for pain relief was started. Oral tapentadol 50 mg was taken twice a day for 7 days, then tramadol 37.5 mg + acetaminophen 325 mg combination was prescribed twice a day. To patients who did not improve their pain with oral mediation, additional intravenous analgesics were given such as ketorolac 30 mg or tramadol 50 mg. Additionally, osteoporosis was evaluated and treated. Within three days after trauma, GRNB was performed under fluoroscopy under local anesthesia.

The sex, age, body mass index (BMI), bone marrow density (BMD), underlying disease, and medications, such as steroids, were analyzed in all patients.

### 2.2. Gray Ramus Nerve Block

The procedure was performed in a biplane angiography room. Patients were placed in a prone position with a soft bed table and arms hanging over the table. Under the fluoroscopic images, a 23 gauge spinal needle was laterally inserted toward the center of the fractured vertebral body. Approximately 3.5~5.0 cm from the midline of the vertebral body just inferior to the transverse process, usually 1.5~2.0 cm lateral to the inferior endplate in case of the thoracic region. The needle was then advanced through just inferior to the pedicle and advanced into a slightly anterosuperior aspect of the foramen. The needle tip was positioned approximately 5–10 mm anterior to the foramen and just above the foramen roof, where the most proximal portion of gray ramus communicans is known to be located [15]. After grossly confirming that the needle was where we wanted it, the correct position was then confirmed radiographically with an injection of 0.5 cc of contrast dye injection (Bonorex). We then injected a combination of 2 cc lidocaine, 5 mg of dexamethasone, and 2 cc of normal saline on each side of the gray ramus communicans of the fractured vertebral body (Figure 2).

During the nerve block procedure, the patient’s vital signs were checked, and after the procedure, we monitored for any signs of complications. (Figure 2 and Figure 3).

### 2.3. Outcome Assessment and Evaluation of Risk Factors for Pain Control Failure

The same questionnaire was administered to all fracture patients, such as VAS (divided into motion and resting score) for pain intensity and characteristics. The Oswestry Low Back Disability (ODI) and Roland–Morris Disability Questionnaire (RDQ) assessments for quality of life were analyzed pre-procedure and at three days, 14 days, 30 days, 90 days, and 180 days post-procedure. The dose of analgesics was monitored for their effect of GRNB before and after the procedure.

The patients were divided into two groups: success and failure groups. We defined ‘treatment failure’ as either the absence of pain improvement or the need for another procedure, such as repeat GRNB or vertebroplasty. Sex, age, smoke, alcoholics, bone marrow density (BMD), BMI, Kummel’s disease, thoracolumbar injury classification and severity score (TLICS), underlying illnesses (e.g., hypertension, diabetics), and medications (e.g., steroids) were analyzed as risk factors of GRNB failure.

### 2.4. Statistical Analysis

Data were analyzed using SPSS (version 20.1, SPSS Inc. Chicago, IL, USA). The primary outcome measures were changes in the VAS, ODI, and RDQ scales over time as determined by using a repeated measure ANOVA. The model included the pattern of covariance between repeated observations to account for correlations between observations within each patient. For the secondary outcome analysis, unpaired Student t-tests or Mann–Whitney tests were used for continuous variables, and chi-square tests or Fisher’s exact test was used for categorical variables. The following baseline characteristics of patients experiencing treatment failure after GRNB were examined for statistically significant differences: The sex, age, BMI, BMD, underlying disease, and medications, such as steroids.

## 3. Results

### 3.1. Participants

Of the 483 patients with vertebral fracture during that period, 94 patients declined the procedure, and 312 patients were excluded due to mild pain of VAS score 3 or less, no acute fractures, or pathological fractures. Finally, 63 patients underwent the gray ramus nerve block procedure (Figure 1).

The characteristics of the 63 study patients are summarized in Table 1. Twenty-four patients were men (24/63, 38%) with a mean age of 66.19 years (±15.95, standard deviation, SD). Forty-three patients had a job (68%), and 21 (33%) had graduated from college. Seven patients (11%) were smokers, and 16 patients (25%) were alcoholics. Twenty-seven patients (43%) were diagnosed with hypertension and eight patients (13%) had diabetes mellitus. Two patients (3%) had Kummel’s disease. Their BMIs were overweight (24.06 ± 3.84), and BMD was in a decreased state (−2.12 ± 1.37). The TLICS results (2.81 ± 1.47) indicate that conservative treatment was recommended.

There were three male and 11 female patients in the treatment failure group. There were 21 male and 28 female patients in the treatment success group. There were no significant differences in education, job, or BMI between the success and failure groups. The TLICS score showed no significant difference between the treatment failure and success groups (3.29 vs. 2.82). There were no significant differences between the two groups, with an average duration of 5 days after fracture. The treatment failure group’s average age was 72.86 years, which was higher than the average age of the treatment success group, 64.26 years (*p* = 0.0088). The treatment failure group’s average BMD was −3.31, which was lower than the average BMD of the treatment success group (−1.79, *p* = 0.0003) (Table 1).

### 3.2. The Changes in Pain and Functional Outcome Overtime after GRNB

Pre-procedure motion VAS was 7.88 ± 0.31. After the procedure, the pain score was serially improved at 3 days (4.94 ± 0.36, *p* < 0.0001), 14 days (4.06 ± 0.35, *p* < 0.0001), 30 days (2.86 ± 0.29, *p* < 0.0001), 90 days (2.39 ± 0.28, *p* < 0.0001). After 90 days, the pain score change did not differ significantly at 180 days (0.27 ± 0.19). The pattern of resting VAS change was different from that of motion VAS. The resting VAS score was 4.37 ± 0.38. Symptom improvement was seen at three days (2.27 ±0.32, *p* < 0.0001), but was then unremarkable at 14 days (1.73 ± 0.29, *p* = 0.24), 30 days (1.24 ± 0.21, *p* = 0.05), 90 days (1.24 ± 0.19, *p* > 0.99) and 180 days (1.06 ± 0.22, *p* = 0.14).

The life quality index, ODI, and RDQ showed improvement over 6 months for all participants. The ODI was improved from pre-procedure (77.95% ± 18.28, standard deviation) to 3 day (60.80% ± 18.65%, *p* =< 0.0001), 14 days (48.41% ± 21.10%, *p* =< 0.0001), 30 days (40.37% ± 20.27%, *p* = 0.00), 90 days (32.49% ± 22.82%, *p* = 0.00) and 180 days (22.73% ± 16.08%, *p* < 0.0001). The initial RDQ, a life quality index, was 19.59 ± 3.57 and improved at 3 days (18.24 ± 3.70, *p* = 0.05), 14 days (15.59 ± 4.34, *p* = 0.00), 30 days (13.35 ± 5.06, *p* < 0.0001), 90 days (10.69 ± 5.92, *p* < 0.0001), and 180 days (9.18 ± 5.45, *p* = 0.00) over 6 months (Table 2 and Table 3).

The frequency of analgesic injection use during hospitalization decreased in the GRNB treatment group. The average number of analgesic administrations decreased from 1.571 pre-GRNB to 0.265 post-GRNB in the success group. However, comparing to failure group, the number of analgesic injections was not significant different (*p* = 0.267).

### 3.3. Risk Factor Analysis

The failure group’s risk factors, such as body mass index, initial thoracolumbar injury classification and severity score, Kummel’s disease, age, bone marrow density (BMD), and underlying disease, were analyzed.

The mean age in the treatment success group was 64.29 years and 72.86 years in the treatment failure group (*p* = 0.0088). The average BMD of the treatment success group was −1.79, which was significantly different from the treatment failure group (−3.31). (*p* = 0.0003). Using a logistic regression model, BMD was a significant risk factor for treatment failure (Table 3).

## 4. Discussion

We report two key results from our study. The first is the time course of pain relief after GRNB. To date, there is no description of a clear change in pain improvement in the literature. In our study, the efficacy of GRNB was illustrated in motion VAS. The improvement started from 3 days immediately after the procedure and lasted until 90 days, with diminishing returns by 180 days. On the other hand, resting VAS was indicative on the 3rd day immediately after the procedure but did not show any significant difference over time. This finding means that GRNB is effective immediately after the procedure, and is more effective in pain caused by movement. Therefore, GNRB may enable earlier walking in fracture patients after the procedure, thereby reducing complications, such as pneumonia and thromboembolic events due to bed rest.

Several papers have demonstrated that GRNB was effective in pain relief in patients with vertebral fractures. In 2001, Chandler et al. in 52 osteoporotic vertebral fracture patients, GRNB after conservative treatment failure was administered with 2% lidocaine and 2% triamcinolone, which was different than the drugs used in our study. During an average follow-up period of 9 months, they observed an improvement in pain in 92% of patients. In 42% of patients, the demand for pain medication decreased, 50% reported high satisfaction, and 25% had moderate satisfaction [12,16]. The change in treatment effect over time and the risk factor of the treatment failure group were not analyzed. Tae et al. showed improvement in pain in 94% of the treatment results for chronic osteoporotic compression fractures which failed in conservative treatment for 4 weeks and the pain relief lasted for four months [15]. In contrast, our study targeted patients with acute vertebral fractures within 3 days after fracture.

In our research, satisfactory pain improvement over three months, without additional treatments, was confirmed in 78% of patients. These were similar results to the study on GRNB in another study, Choi et al [17]. In another study by Kim et al., Radiofrequency (RF) was performed for pain relief in 22 patients with acute vertebral fractures, 48 h after the procedure. The reported pain improved rapidly, and the effect lasted up to 3 months at follow-up, similar to our results [10]. We summarized our result comparing with reference (Table 4) The second finding is that the GRNB effect decreased in the osteoporotic patients. When analyzing the clinical factors in the treatment failure group, there was a significant difference in BMD from the treatment success group (−3.31 in failure versus −1.79 in success, *p* = 0.0038). We think that in osteoporotic vertebral fracture patients, the vertebral body’s stability was decreased, leading to persistent and longtime back pain after the injection. In previous GRNB-related papers, no risk analysis was performed for the patient group with a low treatment effect.

The strength of our study is that the same treatment protocol for acute fractures was applied to all patients. The treatment effect was recorded through the same questionnaire survey at pre-procedure and on the 3rd, 14th, 30th, 90th, and 180th days after the GRNB block, allowing us to evaluate the effect over time. Our study’s other strength is that we define the treatment failure group and analyze the risk factors in this group.

A limitation of our study is that it was a retrospective study, and there was no control group. Compared to conservative treatment alone or shame procedure or cement augmented procedure such as vertebroplasty or kyphoplasty, we may identify the effective time point and period compared to control group. And without a control group, the effect of GRNB may be minimal clinical differences. Another limitation is that the acting time of the drugs used in the block is short acting agent. Various block regimens, such as triamcinolone and RF, were used in other studies. To date, there have been no studies on the difference between these drugs. If a paper that addresses these problems is published in the future, it is believed that a reliable verification of the effect of GRNB treatment will be possible. The other limitation is that, radiologic factors such as changes in the morphology of fractures have not been analyzed as risk factors for treatment failure. The kyphotic angular change of the patient’s body shape due to fractures over time may affect treatment failure.

## 5. Conclusions

The GRNB could relieve acute pain and improve quality of life. The motion pain decreased rapidly for three days and was maintained for 90 days after the procedure. Decreased BMD is a risk factor for GRNB treatment failure.

## Figures and Tables

**Figure 1 medicina-57-00744-f001:**
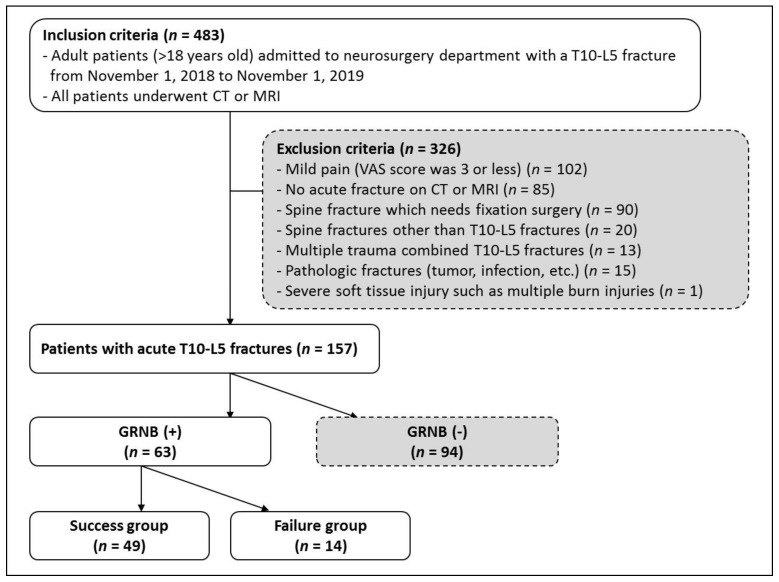
Flow chart of numbers of participants at each stage of study divided into a success group and a failure group. Computer Tomography (CT); Magnetic resonance imaging (MRI); visual analog scale (VAS); gray ramus communicans nerve block (GRNB).

**Figure 2 medicina-57-00744-f002:**
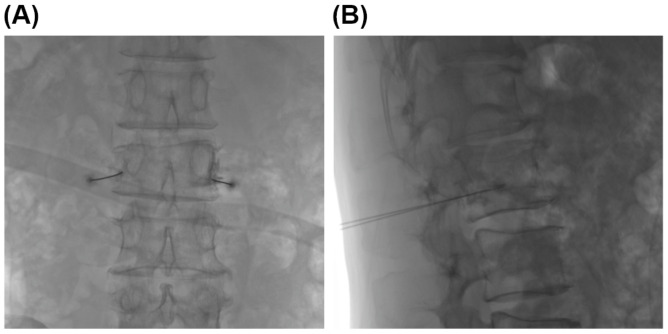
Gray ramus communicans nerve block procedures under the fluoroscopy. Anteroposterior (**A**) and lateral (**B**) fluoroscopic pictures show that it was confirmed through contrast media and images that the tip of the spinal needle is at the position of the gray ramus communicans.

**Figure 3 medicina-57-00744-f003:**
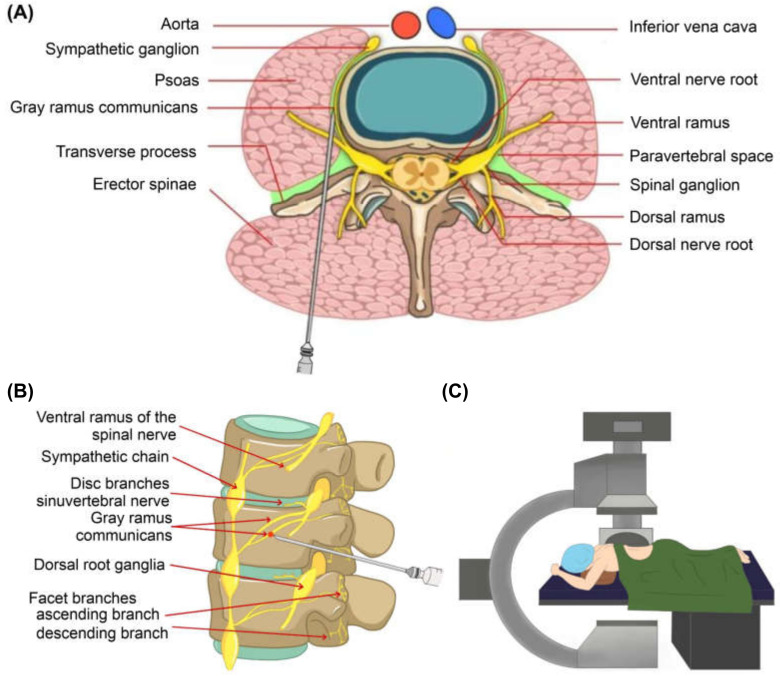
Illustration of gray ramus nerve block procedure: (**A**) axial view, (**B**) sagittal view, (**C**) the setting and position of patient and fluoroscopy for gray ramus communicans nerve block.

**Table 1 medicina-57-00744-t001:** Demographic and clinical characteristics of the study patients (number = 63).

	Total		Failure Group		Success Group		
	*n* = 63		*n* = 14		*n* = 49		*p*-Value
Gender (male)	24	38%	3	21%	21	43%	0.1454
Age	66.19	15.17	72.86	7.77	64.29	16.41	0.0088
Job, yes	20	31%	1	7%	19	39%	0.0267
Post-secondary education	21	33%	2	14%	19	39%	0.0879
Smoke, yes	7	11%	1	7%	6	12%	1.0000
Alcoholism, yes	16	25%	3	21%	13	26%	0.6989
HTN	27	42%	8	57%	19	39%	0. 2207
DM	8	13%	1	7%	7	14%	0.6714
Kummell	2	3%	0	0%	2	4%	1.0000
BMI	24.06	3.61	23.64	2.26	24.18	3.95	0.5275
BMD	−2.12	1.42	−3.31	0.93	−1.79	1.38	0.0003
Duration to GRNB after fracture, days	4.6	4.15	4.79	2.81	4.55	4.51	0.8559
Back pain duration, days	4.04	4.33	3.64	3.15	4.16	4.66	0.6976
TLICS	2.92	3.15	3.29	3.28	2.82	1.48	0.2932

HTN: hypertension, DM: diabetes mellitus, BMI: body mass index, BMD: bone marrow density, GRNB: gray ramus nerve block. TLICS: thoraco-lumbar injury classification and severity score.

**Table 2 medicina-57-00744-t002:** Changes in outcome scales over time after the procedure and comparison of outcome scales at each time point after treatment.

	VAS *n* = 49				VAS at Rest *n* = 49				ODI *n* = 49				RDQ *n* = 49			
Time	Mean	SE	*p*-Value ^a^	*p*-Value ^b^	Mean	SE	*p*-Value ^a^	*p*-Value ^b^	Mean	SE	*p*-Value ^a^	*p*-Value ^b^	Mean	SE	*p*-Value ^a^	*p*-Value ^b^
0	7.88	0.31	<0.0001		4.37	0.38	<0.0001		77.95%	18.28%	<0.0001		19.59	3.57	<0.0001	
3	4.94	0.36			2.27	0.32			60.80%	18.65%			18.24	3.70		
14	4.06	0.35			1.73	0.29			48.41%	21.10%			15.59	4.34		
30	2.86	0.29			1.24	0.21			40.37%	20.27%			13.35	5.06		
90	2.39	0.28			1.24	0.19			32.49%	22.82%			10.69	5.92		
180	2.12	0.31			1.06	0.22			22.73%	16.08%			9.18	5.45		
diff 3 vs. 0	2.94	0.38		<0.0001	2.10	0.40		<0.0001	17.15%	2.58		<0.0001	1.35	0.68		0.05
diff 14 vs. 0	3.82	0.41		<0.0001	2.63	0.42		<0.0001	29.54%	3.96		<0.0001	4.00	0.79		<0.0001
diff 30 vs. 0	5.02	0.41		<0.0001	3.12	0.42		<0.0001	37.58%	3.98		<0.0001	6.25	0.84		<0.0001
diff 90 vs. 0	5.49	0.42		<0.0001	3.12	0.43		<0.0001	45.46%	4.23		<0.0001	8.90	0.91		<0.0001
diff 180 vs. 0	5.76	0.44		<0.0001	3.31	0.45		<0.0001	55.21%	3.46		<0.0001	10.41	0.87		<0.0001
diff 14 vs. 3	0.88	0.42		0.041	0.53	0.45		0.24	12.39%	3.27		<0.0001	2.65	0.73		0.00
diff 30 vs. 3	2.08	0.40		<0.0001	1.02	0.40		0.01	20.43%	3.29		<0.0001	4.90	0.81		<0.0001
diff 90 vs. 3	2.55	0.42		<0.0001	1.02	0.34		0.00	28.31%	3.51		<0.0001	7.55	0.96		<0.0001
diff 180 vs. 3	2.82	0.46		<0.0001	1.20	0.37		0.00	38.06%	2.99		<0.0001	9.06	0.92		<0.0001
diff 30 vs. 14	1.20	0.27		<0.0001	0.49	0.25		0.05	8.04%	2.62		0.00	2.25	0.58		<0.0001
diff 90 vs.14	1.67	0.28		<0.0001	0.49	0.29		0.10	15.92%	2.59		<0.0001	4.90	0.69		<0.0001
diff 180 vs. 14	1.94	0.35		<0.0001	0.67	0.32		0.04	25.67%	2.56		<0.0001	6.41	0.72		<0.0001
diff 90 vs. 30	0.47	0.20		0.02	0.00	0.15		1.00	7.88%	2.41		0.00	2.65	0.53		<0.0001
diff 180 vs. 30	0.74	0.27		0.01	0.18	0.20		0.36	17.63%	2.69		<0.0001	4.16	0.65		<0.0001
diff 180 vs. 90	0.27	0.19		0.17	0.18	0.12		0.14	9.76%	2.19		<0.0001	1.51	0.44		0.00

Values are expressed in mean and standard error. ^a^ Time effect (linear mixed model for longitudinal data). ^b^ Pre-post outcome scale comparison at each time points. SE: Standard error, ODI; Oswestry Low Back Disability, VAS: visual analog scale, RDQ: Roland–Morris Disability Questionnaire.

**Table 3 medicina-57-00744-t003:** Risk factors analysis for pain control failure after gray ramus nerve block.

						Univariate Model				Multivariate Model + Variable Selection			
							95% CI				95% CI		
		Failure Group *n* = 14		Success Group *n* = 49		OR	Lower	Upper	*p*-Value	OR	Lower	Upper	*p*-Value
Gender	M	3	78.57	21	57.14	2.75	0.68	11.11	0.1556				
	F	11	21.43	28	42.86								
Smoke	No	13	92.86	43	87.76								
	Yes	1	7.14	6	12.24	1.81	0.20	16.47	0.5967				
Alcoholism	No	11	78.57	36	73.47								
	Yes	3	21.43	13	26.53	1.32	0.32	5.51	0.6996				
HTN	Yes	8	57.14	19	38.78	0.48	0.14	1.58	0.2257				
DM	Yes	1	7.14	7	14.29	2.17	0.24	19.28	0.4881				
History of osteoporosis	Yes	2	14.29	3	6.12	0.39	0.06	2.61	0.3328				
Steroid use	Yes	0	0.00	0	0.00				-				
Kummell	Yes	0	0.00	2	4.08	1.53	0.04	66.01	0.8256				
Age		72.86	7.77	64.29	16.41	0.96	0.91	1.00	0.0721	0.97	0.94	1.01	0.0972
BMI		23.64	2.35	24.18	3.95	1.04	0.88	1.24	0.6243				
BMD		−3.31	0.97	−1.79	1.38	2.80	1.46	5.39	0.0020	2.67	1.37	5.18	0.0038
TLICS		3.29	1.38	2.82	1.48	0.81	0.54	1.20	0.2907				

HTN: hypertension, DM, diabetes mellitus, BMI: body mass index, BMD, bone marrow density, TLICS: thoracolumbar injury classification and severity score.

**Table 4 medicina-57-00744-t004:** Summarization of our study comparing with References.

	Number	Patient	Intervention Regimen	Intervention	Control	Outcome	Follow Up Period	Complication
Our study	63	acute fracture, within 3 d after fracture (T10~L5)	2% bupivacaine 5 mg Dexamethasone	GNRB	None	VAS, motion 7.88 at pre 4.93 at 3 days 4.61 at 14 days 2.857 at 90 day sResting VAS ODI RDQ	Before And after 3,14, 30, 90, 180 days	No infection No vessel leak
Chandler 2001 [12]	52	After conservative analgesic therapy (TL)	2% lidocaine 2% triamcinolone	GNRB	None	VAS 10 92% at least 1 63% at least 4	Before And after	No report
SW kim 2007 [10]	22	less 2 weeks after trauma (L1-4)	RF	RF	None	VAS 7.8 at pre 2.6 at 48 h 2.8 at 90 d (*p* < 0.005) Modified Macnab 12-6-2-2 at E-G-F-P	At least, 4 month	No significant complication
HS Tae 2003 [15]	36	after failure conservative treatment for 4 weeks or vertebroplasty	2% lidocaine 5 mg Dexamethasone 40 mg Methylprednisolone acetate	GNRB	None	VAS 9.2 at pre 0~3 (80.5%), 4~6 (13,9%) 6< (5.6%) at 24 h0~3 (52.9%) 4~6 (35.3%) 6< (11.8%) at from 4 to 12.5 months.	At least, 4 month	No procedure related complication

GRNB: Gray ramus nerve block, RF: radiofrequency, VAS: visual analog scale, E-G-F-P: excellent-good-fair-poor, TL: thoracolumbar, ODI: Oswestry Low Back Disability, RDQ: Roland–Morris Disability.

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
