# Peer review of "Gray Ramus Communicans Nerve Block for Acute Pain Control in Vertebral Compression Fracture"

_medicina, 2021, doi:10.3390/medicina57080744_

Round 1

Reviewer 1 Report

This paper is a well written retrospective study on the potential effect of gray ramus communicans nerve block for acute pain control in vertebral compression fractures.

The study includes data from 63 patient, who underwent nerve block treatment after vertebral fracture (T10 – L5).

Resting VAS and Motion VAS, as well as ODI, RDO is reported on days 3, 14, 30, 90, and 180.

Patients requiring cement augmentation or repetitive injections were defined as failure. Failure is reported to be about 22% (14 out of 63). Movement pain improvement is reported to be significant after injection within the first the month and quality of life index improved for six months. However, resting pain is reported to be affected by the procedure for only three days. Lower BMD was found to be the only risk facture for failure.

In my opinion, the paper presents an important contribution to the knowledge about treatment options in pain control after vertebra compression fractures.

However, I am unsure about the reported methodology and the conclusions reported from the presented results.

Therefore, this paper needs to be revised to meet its quality potentials and to contribute existing knowledge on treatment options in this field.

My specific comments are below.

Specific Comments:

Line 19: How have the authors defined thoracolumbar vertebral fractures? To my knowledge this term is used for the thoracolumbar junction. By including the whole lumbar spine, fracture patterns may be different due to specific biomechanical characteristics in distinct regions.

Line 56: “The primary goal … was to analyze the time course of pain control after GNRB.” – How did the authors secure that additional pain treatment (e.g. pain medication) was comparable in all 63 patients to distinguish the effect of GNRB?

Line 64: Please consider rephrasing to use the term “thoracolumbar” as already mentioned.

Line 66: I am unclear about the inclusion of all patients older than 18 years. In my opinion the young patients suffering from compression fractures need not to be compared with older patients mainly suffering from compression fractures caused by osteoporosis. Please clarify.

Line 66ff. “…, pathological fractures other than the spine dure to multiple traumas cervical spine combined fracture and those who did not want…” – Please consider rephrasing! I am unclear about the intention of this confusing sentence.

Figure 1:            How could patients decline to participate in a retrospective study?

                        “85, not meet MRI criteria” – I cannot find MRI criteria described in the method section.

“63 patitens were initially assigned GRNB group. They underwent….” How was this done in a retrospective Study?

Line 73: Please define pain treatment other than GRNB. Are there differences? If so, may this have potentially affected the reported outcome?

Line 74: “…three days after trauma, GNRB was performed…” please define the kind of trauma. Are there differences? If so, may this have potentially affected the reported outcome?

Line 133ff: Of the 483 patients with vertebral fracture during that period, 94 patients declined the procedure…” – Why have these patients not used as a control group? I am still unclear about the reported design. Many given information implicate that the study was originally designed to be prospective. The authors are highly encouraged to clarify this issue and to revise the presented methods accordingly.

Table 1: Please name the collumns.

Line 160ff: By reporting the results of the effect of GRNB, I would suggest discussing the values in context with “minimal clinical differences”. This could be taken from literature. Without a control group, it is difficult to rate the effect of this intervention.

Line 189: “The frequency of analgetic drug use …. decreased in the GRNB treatment group” – No other group is reported! How can the authors state on a reduced drug usage?

Was pain medication comparable in the success vs. failure group?

Line 196ff: the increased risk of failure by reduced BMD is an important information. Was there any difference in fracture morphology of fracture progression?

Have non osteoporotic fractures have also been included? This might influence the results and need to be excluded (if possible) or discussed as limitation.

Line 210: “Therefore, GNRB enables earlier walking, in fracture patients after the procedure, thereby reducing complications, such as pneumonia and thromboembolic events due to bed rest” – This statement – which might be true – cannot be concluded from the given results. The authors cannot compare to a control group without having included a control. Please consider rephrasing that effective pain relieve may have these effects.

Line 246: please add that the natural course of pain over time also reduces within 3 – 6 months. No comparison can be given without a pure conservative control group. Additional comparative studies are necessary to reveal the whole effect of GNRB compared to conservative treatment and compared to cemented augmentation techniques.

Line 234: “We think that ….” – Please add that in this patients, other treatment options, like kyphoplasty may be considered – a comparative study between GNRB and vertebra/kyphoplasty might be interesting.

Line 268: How have all patients provided informed consent in a retrospective study design?

Reviewer 2 Report

I salute the authors for their contribution. One needs to have in mind the basivertebral nerve as an important source of endplate and vertebral body-related pain in the thoracolumbar region. And therefore, the conclusions on the gray ramps communicant alone shall be less binding as well.

Author Response

 Title of the Manuscript: Gray ramus communicans nerve block for acute pain control in vertebral compression fracture

Manuscript Number: medicina-1269281

Dear Prof. Dr. Edgaras Stankevičius

We are pleased to submit our revised manuscript " Gray ramus communicans nerve block for acute pain control in vertebral compression fracture" for consideration for publication in Medicina. We have included our point by point responses to reviewer comments below. Changes to the manuscript itself have also been recorded using the track changes function.

We thank you again for considering our manuscript and hope you find it appropriate for publication in Medicina.

Sincerely,

Il Choi, M.D, PhD

Department of Neurological Surgery,

Dongtan Sacred Heart Hospital, College of Medicine, Hallym University

7, Keunjaebong-gil, Hwaseong-si, Gyeonggi-do, 18450, Republic of Korea.

Tel: +82-10-5310-6274

Fax: +82-31-8086-2410

[Point-by-Point Replies to Reviewer’s Comments]

Reviewer 2

We appreciate very much the editor and the reviewers for the constructive comments. We also thank the editor and the reviewers for the effort and time put into the review of the manuscript. Each comment has been carefully considered point by point and responded. Responses to the reviewers and changes in the revised manuscript are as follows.

I salute the authors for their contribution. One needs to have in mind the basivertebral nerve as an important source of endplate and vertebral body-related pain in the thoracolumbar region. And therefore, the conclusions on the gray ramps communicant alone shall be less binding as well.

Authors’ Response: Thank you very much for your thoughtful comment. We searched the literature about Basi-vertebral nerve. (Described by Antonacci et al, J Spinal Disord 1998;11:526–31, S. Becker et al. / The Spine Journal 17 (2017) 218–223) And we found that basivertebral nerve is very import for pain control of thoracolumbar spine. I would keep in mind of the role of basivertebral nerve for spinal origin pain